# *Meso*-Formyl, Vinyl, and Ethynyl Porphyrins—Multipotent Synthons for Obtaining a Diverse Array of Functional Derivatives

**DOI:** 10.3390/molecules28155782

**Published:** 2023-07-31

**Authors:** Vladimir S. Tyurin, Alena O. Shkirdova, Oscar I. Koifman, Ilya A. Zamilatskov

**Affiliations:** 1Frumkin Institute of Physical Chemistry and Electrochemistry, Russian Academy of Sciences, 119071 Moscow, Russia; 2Department of Chemistry and Technology of Macromolecular Compounds, Ivanovo State University of Chemistry and Technology, 153000 Ivanovo, Russia; koifman@isuct.ru

**Keywords:** *meso*-functionalized porphyrin, *meso*-formylporphyrin, *meso*-vinylporphyrin, *meso*-ethynylporphyrin, porphyrin synthon, Vilsmeier–Haack formylation, wittig reaction, schiff base, Heck reaction, Sonogashira reaction, Glaiser coupling

## Abstract

This review presents a strategy for obtaining various functional derivatives of tetrapyrrole compounds based on transformations of unsaturated carbon-oxygen and carbon-carbon bonds of the substituents at the *meso* position (*meso*-formyl, vinyl, and ethynyl porphyrins). First, synthetic approaches to the preparation of these precursors are described. Then diverse pathways for the transformations of the multipotent synthons are discussed, revealing a variety of products of such reactions. The structures, electronic, and optical properties of the compounds obtained by the methods under consideration are analyzed. In addition, there is an overview of the applications of the products obtained. Biomedical use of the compounds is among the most important. Finally, the advantages of using the reviewed synthetic strategy to obtain dyes with targeted properties are highlighted.

## 1. Introduction

The production of porphyrin materials with target-specific structures is based on various methodologies, differing in that natural or synthetic porphyrins are used. Naturally derived porphyrins are substituted at β-pyrrolic position and usually with free-from substitution *meso*-carbons. The synthetic porphyrin core can be easily obtained by tetrapyrrole condensation, and the most popular way is condensation of easily available unsubstituted pyrrole with aldehydes to give *meso*-substituted porphyrins [1]. Thus, two main alternative porphyrin types are used as starting materials in synthesis: (1) β-substituted-*meso*-unsubstituted; and (2) β-unsubsituted-*meso*-substituted (Figure 1). Most of the synthetically obtained porphyrins are *meso*-arylporphyrins, like *meso*-tetraphenylporphyrin (TPP) and *meso*-diphenylporphyrin (DPP). The most popular synthetic porphyrin resembling natural porphyrins is β-octaethylporphyrin (OEP). The starting basic tetrapyrroles then need to be functionalized to impart the required properties to the porphyrin molecule [2,3,4]. The tetrapyrrolic macrocycle can be functionalized by a variety of methods, among which formylation is especially prolific due to the well-developed, very efficient, and easy-to-use formylation methods [5]. The advantage of this functionalization methodology is based on the fact that the aldehyde group is rich in the possibilities of further transformations leading to the addition of functional fragments. The vinyl group is also useful for further functionalizations, and it can be obtained by the Wittig reaction of the formyl-substituted substrate. The acetylenyl group is among the most commonly used for making multichromophore molecules due to its rich coupling possibilities. Thus, formyl, vinyl, and ethynyl groups are magic functions that lead to a variety of products from further transformations, including conversions between them (formyl <=> ethenyl <=> ethynyl). The products obtained through the transformations of these three main synthons are used in a wide variety of applications, including optical sensors [6] and photosensitizers [7,8,9]. The role of β-formyl and β-vinyl-porphyrins in the synthesis of various porphyrin derivatives was summarized in the review [10]. The β-position is more easily accessible to reagents compared to the *meso* position, and there were a wide variety of products synthesized from β-formyl and β-vinyl-porphyrins. The meso position is sterically hindered by the neighboring β-carbon atoms and their substituents, so the reactions of *meso* groups proceed harder. This obstacle is responsible for fewer reported works dealing with *meso* versus β functionalizations and transformations. However, the *meso*-substitution affects the electronic system and spectral properties of the tetrapyrrole macrocycle more strongly. Particularly, the *meso* push-pull substituted porphyrins are the most efficient organic photosensitizers for dye-sensitized solar cells (DSSC) [11]. Thus, it is important to develop *meso* functionalization and transformation methodologies, and the formyl, vinyl, and ethynyl groups present excellent opportunities for primary functionalization and further transformations at the *meso* position.

## 2. *Meso*-Formylporphyrins

Insertion of the formyl group into porphyrins is a primary functionalization of the tetrapyrrole ring, opening opportunities for further transformations including, but not limited to, Wittig [12,13,14], Grignard [14,15,16], McMurry [17], cycloaddition [18], Knoevenagel [19] reactions, and Schiff bases preparation [20]. Ponomarev made a considerable contribution to the chemistry of formylporphyrins and published a corresponding review about 30 years ago, summarizing works reported up to that date [5].

### 2.1. Preparation of Meso-Formylporphyrins

One of the strategies for the preparation of functionalized porphyrins is the utilization of the correspondingly functionalized precursors in the synthesis of the porphyrin core. The formyl group is actively involved in the condensation reaction during tetrapyrrole ring construction, and it needs to be protected. Formylporphyrins were obtained by the usual macrocyclization route to the porphyrins from the masked formyl-containing precursors: 2-formyl-1,3-dithiolane was converted to 5-(1,3-dithian-2-yl)dipyrromethane and mixed MacDonald [2 + 2] condensation with 5-mesytyldipyrromethane and *p-*tolualdehyde led to 5-(1,3-dithian-2-yl)-15-mesytyl-10,20-di(4-tolyl)porphyrin **1** which gave the corresponding 5-formyl-15-mesytyl-10,20-di(4-tolyl)porphyrin **2** after deprotection with DDQ/BF_3_(OEt_2_) (Figure 1) [21].

Similar methods of preparation of formylporphyrins from 1,3-dithiane were proposed by Lindsey [22] and Senge [23]. These methods open opportunities for obtaining 5-, 5,10-, 5,15-, 5,10,15,20-thianyl substituted porphyrins (Figure 2), which can easily be deprotected to the corresponding formylporphyrins with quantitative yield.

The insertion of the formyl group into the already assembled tetrapyrrole macrocycle can be realized both by electrophilic and nucleophilic substitution reactions. 1,3-dithiolane as an umpolung formyl synthon, developed by Seebach and Corey [24], was used to insert formyl via nucleophilic addition of the corresponding lithium salt to the porphyrin. Senge investigated the reaction of the addition of 1,3-dithiane, deprotonated with n-butyllithium, in the presence of *N*,*N*,*N*,*N*-tetramethylethylenediamine (TMEDA), to porphyrins [23]. The products of the nucleophilic addition of the 1,3-dithianyl anion to the 5,15-diphenylporphyrin (DPP) **3**, 5,15,20-triphenylporphyrin (TrPP) **4**, and their nickel complexes (**3**Ni and **4**Ni) were protonated, and the corresponding porphyrinogens were oxidized with DDQ. Subsequent deprotection of the formyl group was carried out with DDQ but in the presence of BF_3_ etherate to give the corresponding *meso*-formyl DPP and TrPP derivatives **5** and **6** (Figure 3). However, low yields and functional group intolerance of the method limit its use.

Takanami developed a more efficient and functional group-tolerant method of nucleophilic formyl group insertion using 2-(trimethylsilyl)pyridine. The reaction proceeded via nucleophilic addition of (2-pyridylmethylsilyl)lithium to DPP **3**, followed by protonation of the intermediate anion and oxidation of the porphyrinogen back to the aromatic porphyrin ring to give DPP-CHO **5** (Figure 4) [25,26]. It should be noted that mild oxidation of the *meso*-silyl porphyrinogen by air led to the *meso*-(hydroxymethyl)porphyrins **7** [27].

The reaction proceeds under mild conditions, and it is applicable to aryl- and alkyl-substituted porphyrins as well as their metal complexes. In addition to monoformyl derivatives, this method makes it possible to obtain diformyl derivatives, as well as more complex aldehydes with high yields. To confirm the mechanism, reactions were carried out with the addition of various electrophiles, such as acyl chloride, methyl chloroformate, isocyanates, and enones, resulting in the corresponding formyl derivatives **8**, variously substituted at the opposite formyl *meso* position [28]. 

Osuka obtained porphyrin Grignard reagents for the first time using metal-iodine exchange between *meso*-iodoporphyrins and iPrMgCl. The utilization of the *meso*-magnesiumporphyrins in reaction with DMF led to the formation of the *meso*-formyl derivatives (Figure 5) [29].

Reactions of electrophilic substitution have found the greatest application in the chemistry of porphyrins as electron-rich aromatics. More reactive *meso* positions are usually attacked by electrophiles. The electrophilic formylation can easily be performed using the Vilsmeier–Haack reaction. The synthesis of *meso*-formyl-β-octaalkylporphyrins using Vilsmeier–Haack formylation was first reported more than half a century ago [30,31]. To date, this reaction has become one of the most popular and efficient for obtaining *meso*-formylporphyrins.

Porphyrin metal complexes resistant to HCl, released during the reaction, are substrates for formylate. It is known that the rate of formylation decreases with increasing electron acceptor ability of the metal cation in the series M(II) > M(III) > M(IV) and among M(II) in the series Ni(II) > Cu(II) > Pd(II) > Pt(II). Of the numerous variants of the reaction, the Vilsmeier reagent made from DMF/POCl_3_ is usually used in porphyrin chemistry. The mechanism of the reaction is as follows: at the first stage, the electrophilic Vilsmeier reagent attacks the nucleophilic *meso* position of porphyrin, resulting in the formation of the iminium salt, which is the so-called “phosphorus complex”. Subsequent treatment of the phosphorus complex with water leads to the hydrolysis of the iminium salt, resulting in the formation of formylporphyrin (Figure 6).

### 2.2. Reactions of Meso-Formylporphyrins with Nitrogen Nucleophiles

Ponomarev investigated the Vilsmeier–Haack formylation of a number of Cu(II), Ni(II), and Pd(II) complexes of β-octaalkylporphyrins and chlorins and the subsequent transformations of the *meso*-formyl derivatives [5,32]. As an electrophile, the formyl group can react with a variety of nucleophiles, including organometallic reagents, CH-acids, heteroatom nucleophiles (amines, thiols), electron-rich aromatic cycles, and heterocycles. The reaction of formylporphyrins with amines leads to azomethines or Schiff bases, and the corresponding studies up to the mid-1990s were summarized in the review of Ponomarev [20]. *Meso*-formyl porphyrins react with amines to give the corresponding imines; hydroxylamines give oximes; and hydrazines produce hydrazones. The Vilsmeier–Haack formylation of β-octaalkylporphyrin **13** produces the stable phosphorus complex **14** due to the sterical hindrances retarding hydrolysis of the complex. Ponomarev isolated phosphorus complexes of various porphyrins and used them instead of formylporphyrins for the preparation of Schiff bases [20]. The iminium group of the phosphorus complex is more active to nucleophilic attack compared to the formyl group. Azomethine derivatives of nickel and palladium complexes of various porphyrinoids, including OEP, tetraalkyl esters of coproporphyrins I and II, mesoporphyrin IX, and mesochlorin e6, were obtained by direct interaction of “phosphorus complexes” with amines (Figure 7) [33,34,35]. 

Azomethine substitution at the *meso* position noticeably shifts the UV-Vis absorption spectra of porphyrins to long wavelengths, thus making their metal complexes potential photosensitizers [34]. Schiff bases also impart basic properties to the porphyrins, and the corresponding Pt(II) and Pd(II) complexes of azomethine derivatives of OEP and tetramethyl coproporphyrin I were investigated as sensor dyes for measuring proton and oxygen concentrations using an optical noninvasive method [36,37]. The corresponding phosphorescent probe, based on the Pt(II) complex of *meso-*(*N*-methylimino)-OEP for cellular diagnostics using dual oxygen and pH measurements in living cells, has been reported [38].

The imino group, which can be obtained by interacting the formyl group with an amine, was used as a linker to bond two tetrapyrrole chromophores into a dyad. The interaction of Zn(II) complexes of *meso*-formyltriphenylporphyrin **6**Zn and *meso*-aminotriphenylporphyrin **17**, catalyzed by Lewis acid ZnBr_2_, led to the corresponding dimer of TrPP **18** (Figure 8) [39]. The TrPP macrocycles are not coplanar in dimer **18** and consequently not conjugated. Nevertheless, there is some interchromophore communication, and dimer **18** features increased two-photon absorption, which can be used in PDT, providing deeper and more targeted treatment [40].

Schiff bases are useful synthons, as they can be subjected to further transformations. Elimination of *meso*-oximes led to *meso*-cyanoporphyrins [41]. Intramolecular cyclization was observed when *meso*-oxime **19**Zn was vigorously stirred for a few hours in methylene chloride with a small amount of water. The probable intermediate chlorin with the fused 1,2-oxazin ring underwent hydroxylation via a peroxide mechanism to form stable hydroxychlorin **20**Zn. The use of an oxidant, lead tetraacetate, gave the product the same fused 1,2-oxazin but with a vinyl group instead of ethyl and hydroxyl (Figure 9) [42,43]. Apparently, the difference between products **20** and **21** is in the water molecule, which was probably eliminated in the case of **21**. The UV-Vis spectra of the 1,2-oxazin annulated porphyrinoids **20** and **21** possess strong absorption bands in the red region.

Treatment of the Ni(II) complex of *meso*-(*N*-methylimino)-OEP **15**Ni with t-BuOK led to the formation of the corresponding *meso*-nitrile **22**, *meso*-amide **23**, and *meso*-hydroxy **24** derivatives (Figure 10). The latter was demetalated with sulfuric acid, resulting in phlorin **25** with strong light absorption in the region of 700 nm [35].

Thermolysis of *meso*-alkylimines of β-substituted metal porphyrins led to the formation of cyclopentane-fused derivatives (Figure 11) [44,45,46]. The *meso*-imines of the tetraalkyl ester of coproporphyrin I (**26** and **27**) were transformed into a mixture of cyclopentane and cyclopentane-lactam bicycle fused derivatives (**28**–**30**) [35].

*Meso*-hydrazones of nickel and palladium complexes of OEP and coproporphyrin I ethyl esters were obtained as a mixture of *E*- and *Z*-isomers by the reaction of the corresponding *meso*-formyl derivatives with hydrazines catalyzed by trifluoroacetic acid (Figure 12) [47]. 

*N*-tosylhydrazones **31** and **33** reacted with bases, generating an in situ porphyrin derivative of diazomethane, which released nitrogen molecules to give *meso*-carbene derivatives of porphyrins. Subsequent intramolecular insertion of the carbene into the CH bond of the neighboring β-substituent led to the corresponding fused cyclopentane (**32** and **34**) [48]. The cyclopentane fused products obtained were the same as in the thermolysis of azomethine **26**; however, the second product in the carbene-based reaction of **33** was cyclohexane fused product **35** instead of bicyclic lactam **29** obtained in the thermolysis of azomethine **26** [35], and yields of the products in the carbene-based cyclization were appreciably higher compared to the thermolysis (Figure 13).

Unsubstituted *meso*-hydrazones of OEP **36** and also β-octaethylchlorin (OEC) were used in the preparation of dyads **37** and **38** bridged with the azine group [49]. Derivatives of the natural chlorins, methyl pyropheophorbide-*a* (PPPa) and methyl pyropheophorbide-*d* (PPPd), reacted with the *meso*-hydrazones of OEP and OEC, leading to the formation of the corresponding porphyrin-chlorin and chlorin-chlorin dyads (Figure 14) [49]. Upon irradiation of the dyads, the energy of the excited state was efficiently transferred from the OEP (OEC) components to the pyropheophorbide chromophore. However, the chromophores weakly interacted in the ground state; therefore, the azine group was regarded as a conjugation switch, usually in the off state but capable of being turned on with a sufficiently strong driving force.

One-pot *meso*-formylation, hydrazone, and azine formation were performed for *meso*-(trifluoroacetamido)-OEP **39** [50]. Under the conditions of formylation, the amide group was unexpectedly partially oxidized to form hydroxamic acid **42** (Figure 15). The combined influence of trifluoroacetamide and arylazine groups in the products **41**–**43** led to strongly increased absorption near 500 nm and considerably red-shifted Q-bands up to 650 nm. Azine-bridged porphyrin-chlorin dyad **44** was obtained from *meso*-(trifluoroacetamido)-OEP **40a** and PPPd (Figure 15). The dyad features substantial growth in the Q-band intensity as well as a red-shifting Soret band compared to the similar dyad without the trifluoroacetamido substituent.

The *meso-*formyl group of porphyrins can be transformed to azomethine ylide by interaction with *N*-methylglycine. 1,3-dipolar cycloaddition of the intermediate azomethine ylide to the double bond leads to porphyrins with *meso*-fused heterocycles. The porphyrin—fullerene conjugate **46** was obtained this way from *meso*-formyltriarylporphyrin **45**, *N*-methylglycine, and C_60_ fullerene (Figure 16) [18]. The irradiation of the dyad led to the formation of the exciplex due to the strong interaction between the porphyrin and C_60_ chromophores at short distances.

Porphyrins functionalized with *meso*-fused 2-imidazolyl heterocycles were synthesized from the 5-formyl-10,20-diarylporphyrins and phenanthrene- or phenanthroline-5,6-dione in the presence of ammonium acetate (Figure 17) [51]. The ruthenium phenanthroline complex of the free base porphyrin-imidazo[4,5-f]phenanthroline conjugate showed good binding ability to DNA and was capable of DNA photocleavage, which allows us to regard the complex as a potential photosensitizer for PDT [52].

### 2.3. Reactions of Meso-Formylporphyrins with Miscellaneous Nucleophiles

A family of push-pull quinoidal porphyrins was obtained from a *meso*-formyl porphyrin **48** through the attachment of 1,3-dithiolane (benzo-1,3-dithiolane) and malononitrile fragments at the opposite *meso* positions of the 5,15-diarylporphyrin (Figure 18) [53].

Directly linked porphyrin-corrole dyads **52a**–**c** were formed during condensation of the *meso*-formyltriarylporphyrin **51** with dipyrromethane (Figure 19) [54]. A similar corrole-porphyrin-corrole triad was obtained when the 5,15-bisformylporphyrin was placed into the reaction. The strong exciton coupling between closely placed chromophores and reversible energy transfer were shown to exist in the dyad [55]. Directly *meso-meso*-linked porphyrin dimers and oligomers were obtained using condensation of *meso*-formylated porphyrins with pyrrole [21]. Such porphyrin dimers and oligomers were shown to act as prospective photosensitizers [56].

### 2.4. Reactions of Meso-Formylporphyrins with Organometallic Reagents

Among organometallic reagents, Grignard reagents were usually employed in the reactions for *meso*-formylporphyrins. Alkyllithium reagents gave the expected products of addition to the carbonyl group only with less hindered β-formyl derivatives [15]. Grignard reagents interact with *meso*-formylporphyrins, leading to the formation of the corresponding secondary alcohols, but due to steric factors and perhaps other causes, this reaction proceeds somewhat slowly. Especially retarding is the presence of β-alkyl substituents. For example, Ponomarev reported the formation of the Mg complex of the formylporphyrin without significant formation of the target *meso*-(1-hydroxyethyl)-OEP upon treatment of *meso*-formyl-OEP (OEP-CHO) with MeMgI under heating [57]. Smith carried out a similar reaction with a free base of OEP-CHO and a Zn(II) complex of OEP-CHO (ZnOEP-CHO), which resulted in 15-alkylated products, and the formyl group remained intact (Figure 20) [58]. However, when Johnson and Arnold used the Ni(II) complex of OEP-CHO in the same reaction, Ni(II) 5-(1-hydroxyethyl)-OEP was obtained, as expected [16]. Water was easily eliminated, yielding Ni(II) 5-vinyl-OEP **54** (Figure 21). The Wittig reaction is cleaner and more efficient, as well as tolerating various functional groups such as esters. Various Wittig regents interacted with the *meso*-formyl group of β-substituted porphyrins to form *meso*-vinyl **54** (Figure 21), 2-(ethoxycarbonyl)ethenyl **59**, 2-cyanoethenyl **56** (Figure 22), and other alkenyl groups [5,14]. *meso*-formylporphyrins with *meso*-aryl groups without β-substituents were transformed to the corresponding *meso*-vinylporphyrins [59]. The products of the Wittig alkenylation can further be cyclized; for example, the methyl ester of coproporphyrin I was converted to the corresponding derivative of copropurpurin I **60** (Figure 22). Purpurins and benzochlorins have an additional annealed cycle through *meso* positions and β-positions, which affect the π-electron system, leading to a bathochromic shift of the absorption bands. These annelated porphyrins and chlorins are more stable compared to other chlorins and have comparable electron-optical properties suitable for PDT. They have a higher efficiency than *meso* tetraphenylporphyrin and higher absorption in the longer wavelength region. The benzochlorins were shown to be of low dark toxicity towards Chinese Hamster ovary cells, whereas in the presence of light, total cell killing was observed at concentrations of the photosensitizer below 1 μg/mL [60]. These promising properties of the annelated porphyrin derivatives attract the attention of medical researchers [61]. One of the representatives of this type of compound, tin etiopurpurin complex **64**, was used as a photosensitizer for PDT in human clinical trials. The drug was obtained from nickel etioporphyrin, which was formylated and reacted with a Wittig reagent, yielding *meso*-acrylate derivative **62**, which was cyclized in acid to give etiopurpurin **63**. The latter was metalated with tin(IV) chloride to give the drug for PDT (Figure 23) [62]. The selectivity for the cyclization proceeding exclusively towards the carbon carrying the ethyl group vs. the carbon carrying the methyl group.

There are several published papers devoted to the synthesis of benzochlorin derivatives based on octaethylporphyrin and hematoporphyrin IX, as well as the study of their properties as photosensitizers [63,64,65,66]. In particular, the preparation of variously substituted benzochlorins containing fluorinated or alkyl groups has been reported [64].

The formyl group can be transformed to the 2-haloethenyl group in one step using two different reactions. The Wittig reaction of NiOEP-CHO **53** with bromomethyltriphenylphorphonium bromide led to *meso*-(2-bromoethynyl)NiOEP **65** as a major (*E*)-isomer with a 55% yield [67]. However, the side metal-halogen exchange reaction led to the formation of lithiated methylene ylide and subsequently the formation of *meso*-vinyl byproduct **54**, which was hard to separate. The use of potassium t-butoxide in THF avoided contamination and produced a 53% yield (Figure 24) [68]. Alternatively, the Takai reaction with iodoform catalyzed by CrCl_2_ led to the formation of *meso*-(2-iodoethynyl)porphyrin **66** (Figure 25). The obtained 2-haloethenyl derivatives were used as substrates of the cross-coupling reactions and precursors for the preparation of *meso*-ethynylporphyrins.

Transformation of the formylporphyrins into dimers bonded with an ethene bridge can be performed using low-valent titanium, which is called the McMurry reaction [17]. Cu(II) and Ni(II) complexes of OEP-CHO were dimerized under the action of TiCl_3_ and Zn/Cu to form the corresponding complexes of dimers linked with the ethylene bridge in the form of a mixture of *cis* and *trans* isomers (Figure 26) [69].

The similar dimerization of the Ni(II) *meso*-formyltriarylporphyrin **45** Ni was observed as a side reaction of coupling with tetraphenylzirconacyclopentadiene in the presence of AlCl_3_, along with products of cross-coupling: porphyrin—cyclopentene **69**, **70**, and cyclopentadiene **68** hybrids (Figure 27) [70].

### 2.5. The Reaction of Meso-Formylporphyrins with CH Acids

The Knoevenagel reaction allows for the transformation of formylporphyrins into the corresponding acrylic acid derivatives. The CH acid nucleophiles were introduced into the reaction with *meso*-formylporphyrins, leading to the formation of substituted *meso*-ethenyl porphyrin derivatives. *Meso-*formyl-diarylporphyrin **71** reacted with nitromethane, dimethylmalonate, and malononitrile in a mixture of piperidine, acetic acid, and toluene, leading to the corresponding substituted *meso*-(2-nitroethylene) **72** and *meso*-methylenemalonate **73** derivatives (Figure 28) [71]. The *meso*-cyanoacrylate derivative **74** of Zn(II) *meso*-formyl-triarylporphyrin was obtained by heating in a mixture of piperidine and methanol for 16 h [72]. The product **75** of the reaction of *meso*-formyldiarylporphyrin with malononitrile containing a *meso-*dicyanovinyl group was shown to act as a fluorescence ‘‘turn-on’’ cyanide probe [73]. The *meso*-nitroethylene derivative **72** was utilized as fluorescence turn-on probes for biothiols as it exhibited fast fluorescence enhancement and high selectivity towards thiols based on the Michael addition mechanism [71]. It was also successfully applied to fluorescent cell imaging in the NIR wavelength range.

NiOEP-CHO **53** was much less reactive in the Knoevenagel reaction compared to β-unsubstituted porphyrins due to the sterical hindrances and was gradually degraded under basic reaction conditions. In order to activate the formyl group against the attack of CH acid nucleophiles, Lewis acid TiCl_4_ was used in pyridine. The Lewis acid promoted the Knoevenagel reaction of the NiOEP-CHO **53** with malonic ester and heterocyclic CH acids [19]. The heterocyclic derivatives of porphyrins **77**–**79** linked with an exocyclic C=C double bond were obtained both by cyclization of the Knoevenagel product **76** from the reaction with malonic ester (Figure 29) and by the Knoevenagel condensation of formylporphyrin with thiohydantoin and thiobarbituric acid (Figure 30) [74].

The UV-Vis spectra of the heterocyclic conjugates contain new bands that arose from the interaction of the conjugated chromophores as well as bathochromically shifted original absorption bands. Particularly dramatic changes were observed in the UV-Vis spectrum of the porphyrin conjugate with thiobarbituric acid, which exhibited substantial absorption enhancement in the visible spectral range due to the considerable π-electron conjugation between tetrapyrrole and heterocyclic chromophores.

To sum up, *meso*-functionalization of porphyrins with the formyl group provides a powerful tool for the development of diverse porphyrin derivatives possessing valuable properties. In particular, promising photosensitizers with strong, red-shifted absorption bands, including NIR bands, were obtained from the *meso*-formyl porphyrins via the formation of annulated cycles such as benzochlorins and dibenzobacteriochlorins. *Meso*-imino derivatives were applied as sensor dyes in the multi-modal, multi-analyte optochemical sensing platform for cell diagnostics. Easily formed with the help of the formyl group, porphyrin conjugates with heterocycles can be used as biologically active compounds and in sensing applications. Imino- and azino-bridges represent two alternatives for bonding porphyrins into dyads, utilizing various pathways for energy transfer between the chromophores. Currently, the post-derivatization of the *meso*-formylporphyrins is under intense development.

## 3. *Meso*-Vinylporphyrins

Vinyl-substituted porphyrins are direct derivatives of formyl porphyrins, usually being obtained from the latter via the Wittig reaction. The vinyl group is a versatile nucleophilic synthon complementing electrophilic formyl. The electrophilic addition/substitution reactions and the modern catalytic cross-coupling and direct CH-functionalization methods are inherent to the vinyl group. The vinyl substituent can further be converted to the ethynyl group.

### 3.1. Preparation of Meso-Vinylporphyrins

*Meso*-vinyl substituted porphyrins can be obtained from the *meso*-formylporphyrins using the Grignard and Wittig reactions as described in a previous section [14,16]. However, the efficient Vilsmeier–Haack porphyrin formylation reaction is limited to certain metal complexes and cannot be used for free bases or more labile zinc complexes. The alternative formyl preparation methods are based on palladium-catalyzed cross-coupling reactions, which are tolerant to other functional groups but require primary halogenation of the porphyrin core. Heck and Stille reactions with *meso*-bromoporphyrins led to the corresponding *meso*-alkenylporphyrins [59]. Starting *meso*-bromo derivatives can easily be obtained by bromination of *meso*-di(tri)arylporphyrins with NBS [75,76]. *Meso*-vinylporphyrins were synthesized from *meso*-bromoporphyrins by the Pd-catalyzed Stille reaction with vinyltributyltin (Figure 31) [59,75,77].

### 3.2. Transformations of Meso-Vinylporphyrins

The carbon-carbon double bond of the vinyl group is susceptible to electrophilic addition. However, bromination of the *meso*-vinyl-NiOEP **54** with pyridinium tribromide led to the product of electrophilic substitution: the 2-bromoethenyl derivative **65** as a mixture of (*E*) and (*Z*) isomers (Figure 32) [16]. Bromination of *meso*-vinyl-TrPP proceeded similarly [78]. Possibly, the sterical hindrances at the *meso* position hamper the second bromine atom addition. The bromination product *meso*-(2-bromoethenyl)porphyrin **65** can serve as a substrate in palladium-catalyzed cross-coupling reactions and as a precursor of *meso*-ethynylporphyrin. Cross-coupling of iodo derivatives proceeds easily, and this was the reason for the exchange of bromine for iodine via palladation/iodination (Figure 33). The subsequent Suzuki coupling of the *meso*-(2-iodoethenyl)-TrPP **82** with *meso*-pinacolboronyl-TrPP **83** led to the TrPP dimer **84** being bridged by an ethene linker (Figure 34) [78]. The dimer **84** exists in solution in a number of conformations differing in dihedral angles between the porphyrin and alkene planes. More coplanar conformers have appreciable π-electron conjugation and interchromophore interaction across the bridge.

The *meso*-vinyl group participated in electrophilic substitution reactions not only during bromination but also during formylation in Vilsmeier–Haack conditions. *Meso*-vinyl-NiOEP **54** was transformed to the corresponding *meso*-acroleinic derivative **85** by treatment with the Vilsmeier reagent DMF/POCl_3_ in 1,2-dichloroethane (Figure 35) [79]. This compound can also be obtained using a short method by vinylogous formylation of NiOEP **13**Ni with *N*,*N*-dimethylaminoacroleine/POCl_3_ [69]. The treatment of the *meso*-acroleine derivative **85** with concentrated sulfuric acid led to cyclization involving ethyl migration to give benzochlorin **86** (Figure 36) [69,79].

The octaethylbenzochlorin possesses significant red-light absorption, which makes it a potential photosensitizer for PDT [8,9,80,81]. Some benzochlorin derivatives can cause significant tumor regression at doses as low as 0.5 mg/kg body weight [82]. The product of the double cyclization of *meso*-bis-acroleinyl-OEP **87** is dibenzobacteriochlorin **89** [69], which possesses a strong absorption band in the region of 752 nm, thus fully corresponding to the tissue transparency window (Figure 37) [8,82]. The conjugates of similar benzochlorin with carbohydrates were screened using the galectin-binding-ability assay and exhibited an enhancement of about 300–400-fold compared to lactose. All conjugates were also shown to possess good photosensitizing efficacy with fibrosarcoma tumor cells [65].

Arnold studied the functionalization of porphyrins using the Heck reaction [59]. The Heck coupling of 5-vinyl-10,15,20-triphenylporphyrinatonickel(II) **80**Ni with 50 eq. iodobenzene was performed using a 20 mol% Pd(OAc)_2_ catalyst with triphenylphosphine ligand K_2_CO_3_ as a base in a mixture of DMF and toluene heated to 105 °C for 48 h. As a result of the reaction, two major regioisomers were formed: *trans*-1,2-disubstituted ethene **90** with a 54% yield and 1,1-substituted product **91** with a 20% yield. It should be noted that 1,1-substitution with a very bulky porphyrin substituent is not usual in the Heck reaction. However, an even more curious result was obtained in the reaction with 5-vinyl-10,20-diphenylporphyrinatonickel(II) **92**Ni, where additionally the β-substituted *E*-(2-phenylethenyl)porphyrin **94b** was formed (Figure 38).

2-subsituted ethenyl porphyrins were alternatively obtained using the Heck coupling of the *meso*-bromoporphyrins **95** and **97** with substituted ethene (usually with an electron acceptor group) [59]. The Heck coupling of 5-bromo-TrPP **95** and 5,15-dibromo-10,20-diarylporphyrin **97**, as well as their Ni(II) and Zn(II) complexes with large excesses of methyl acrylate, styrene, and acrylonitrile, led to the corresponding mono- and dialkenyl functionalized porphyrins (Figure 39). *E*-isomers were obtained predominantly, but some amount of *Z*-isomer was also formed in the case of less sterically demanded acrylonitrile. Partial debromination was observed during the reaction. The free-base porphyrins were partially metalated with palladium. Zinc complexes were less stable compared to nickel complexes and were slightly demetalated and transmetalated.

The Heck reaction was used to produce a porphyrin dimer bound by ethene. However, a large excess of one substrate over the other cannot be used in coupling two porphyrin substrates, as was used in reactions with small molecules, because both coupling compounds are very precious. Consequently, the reaction was too slow, side reactions rose, and debromination of the bromoporphyrins occurred predominantly. The coupling of *meso*-bromo-TrPP **95** with *meso*-vinyldiarylporphyrinatonickel **92**Ni did not give the target *meso*-ethenyl-linked dimer but rather *meso*, β-ethenyl-linked dyad **102** (Figure 40). Free-base bromoporphyrin, Ni(II), and Zn(II) complexes gave the corresponding dyad yields of 23, 33, and 15%, respectively. The electronic absorption spectra of the dyads revealed a modest degree of interchromophore interaction via a partially conjugated bridge. This was explained by two factors: twisting of the ethene bridge at *meso* position with respect to the tetrapyrrole plane reduces π–π conjugation, and linkage through the β-carbon has smaller orbital coefficients and consequently less influence on the π–electron system. The *meso-meso*-linked *meso*-arylporphyrin dimers were obtained by several other transition metal-mediated methods: the Suzuki reaction of the *meso*-(2-iodoethenyl)porphyrin with *meso*-pinacolboronylporphyrin, described above [78]; the Stille coupling of 1,2-di(tributyltin)ethene with *meso*-bromoporphyrin [83]; *meso*-iodoporphyrin [39]; and the McMurry coupling of *meso*-formylporphyrin, described above in the formyl section [84].

The *meso-*vinyl group can also be functionalized via catalytic direct CH-functionalization reactions. The direct C-H borylation of the *meso*-vinyl group in NiOEP **54** was performed with Cu(II) complex as a catalyst, yielding the *meso*-(2-pinacolboronylethenyl)porphyrin **103**Ni, which was shown to act as a nucleophilic partner in the Suzuki cross-coupling leading to porphyrin derivatives **104** and **105** with an extended π-conjugation through the carbon double bond [85]. The oxidative homocoupling of the borylporphyrin **103**Pd produced the dimer **106** (Figure 41) [86]. Thus, this strategy of *meso*-vinyl transformations allows for the attachment of various chromophores through the unsaturated bridges. The products of couplings possess some degree of conjugation across the bridge and interchromophore interaction, which induces a bathochromic shift of absorption bands.

The *meso*-vinyl group in porphyrins differs in properties from β-vinyl and other vinyl-substituted aromatics. The sterical hindrances at the *meso* position decrease the reactivity of the vinyl group and change the results of reactions. For example, interaction with electrophiles led to electrophilic substitution instead of addition, like in aromatics. The Heck reactions proceeded harder. Probably, this was one of the reasons for quite a small amount of work devoted to the transformations of *meso*-vinyl groups, especially compared to the β-analogs. The rich potential of the *meso*-vinyl function is to be revealed.

## 4. *Meso*-Ethynylporphyrins

Due to the absence of the sterically interacting extra substituents at the linking carbon, the *meso*-acetylenyl group is coplanar with the macrocycle ring and fully π-electronically conjugated to the tetrapyrrole aromatic system in contrast to other *meso*-attached unsaturated groups like vinyl, phenyl, etc. [87,88,89,90,91]. The acetylene linker has been shown to allow efficient π-conjugation and strong electronic communication between chromophores [92,93,94]. This advantage of the triple bond linker is used when one needs to create extended conjugated systems.

### Synthesis of Meso-Ethynylporphyrins

There are several ways to obtain *meso*-acetylenylporphyrins, including classical functional group transformations and modern catalytic cross-coupling reactions. The oldest route utilized alkynyl-substituted precursors in the assembly of the porphyrin core. MacDonald [2 + 2] condensation of dipyrromethane with trimethylsilylpropynal led to the 5,15-bis(trimethylsilylethynyl)porphyrin **107** in 11% yield, which was deprotected and converted to the Ni(II) 5,15-bisacetylenylporphine **108** (Figure 42A) [95]. In some cases, the product of the reduction of one triple bond can occur. 5-alkenyl-15-alkynyl-porphyrin **108** and 5,15-dialkynyl-porphyrin **109** were formed selectively depending on the choice of solvent (Figure 42B) [96]. The alkenyl group arises from a protonation followed by intramolecular 1,2-hydride transfer from the methine position of porphyrinogen [96].

The classical way to obtain alkyne is through the elimination of hydrogen halide from halo-vinyl. The first *meso*-(2-bromoethenyl)-NiOEP **65** was obtained using the Wittig reaction of NiOEP-CHO **53** with bromomethyltriphenylphosphonium bromide. Then the Wittig product **65** was treated with dimsyl sodium, yielding *meso*-acetylenyl-NiOEP **110**Ni with an 86% yield (Figure 43) [95,97].

The most common way to insert an acetylenyl group into porphyrins is based on the Sonogashira reaction [97]. However, the Sonogashira reaction can be accompanied by some side reactions. The most common complication is the oxidative dimerization of the terminal alkynes [98]. Trialkylsilyl-protected acetylenes are often used, like trimethylsilyl- and triisopropylsilylacetylene, instead of gaseous acetylene, so the products are not able to dimerize. *Meso*-acetylenylporphyrin **112** was prepared in 82% yield by the Sonogashira coupling of *meso*-bromo-porphyrin **111** with a 2.5 eq. excess of triisopropylsilylacetylene catalyzed by 20 mol% Pd(PPh_3_)_2_Cl_2_ and 3 eq. CuI in THF with triethylamine (Figure 44) [99].

When the less bulky trimethylsilyl protection group instead of triisopropylsilyl was used in the Sonogashira reaction of 5-iodo-10,15,20-tris(3,5-di(*tert*-butyl))porphyrin **113** with an excess of trimethylsilylacetylene, the byproduct of the addition of the second acetylene molecule **114b** to the triple bond was obtained (Figure 45) [100].

The most popular transformation of the *meso*-acetylenylporphyrins is also the Sonogashira coupling, leading to the triple bond linked dyad of the porphyrin with another fragment. Linking electron donors or acceptors to the porphyrin ring through the C≡C triple bond significantly affects the tetrapyrrole aromatic system. For example, to attach the salicylic acid anchor to the DPP **3**, the latter was first brominated at the *meso* position followed by zinc metalation and catalytic coupling with triisopropylsilylacetylene. Metalated porphyrins are usually applied as substrates in transition metal-catalyzed reactions instead of free bases to prevent scavenging of the catalytic metal by coordination with the macrocycle. Silyl protection is removed with TBAF, and the *meso*-acetylenyl porphyrin was next coupled with an iodo derivative of the salicylic acid, yielding the product **115**, with the anchoring group being conjugated to the porphyrin through the triple bond (Figure 46) [99].

Strong electronic communication was observed between the triarylamine donors and porphyrin ring in the compound obtained by the Sonogashira reaction of *meso*-diacetylenylporphyrin with iodophenyldiarylamines (Figure 47) [101]. The UV-Vis absorption spectra are considerably bathochromically shifted relative to the starting *meso*-arylporphyrin and exhibit a broad Soret band and an intense Q band.

With the help of Sonogashira coupling, a new photochromic porphyrin-perinaphthothioindigo dyad **117** was prepared (Figure 48). One of the iodine atoms of the diiodo-perinaphthothioindigo reagent was reduced during the reaction, which led to the predominantly mono-substituted product. A small amount of the bisporphyrin-substituted triad was also obtained. Due to the extended conjugated system, the dyad **117** exhibited efficient two-photon absorption properties and clear photochromic switching between *cis* and *trans* isomers [102]. The two photon absorption cross-section maxima for both isomers appeared around 850 nm, with values of 2000 GM for *trans* and 700 GM for *cis* isomers.

The dyad **119** of PtOEP with di(*p*-acetylenylphenyl)anthracene was obtained using Sonogashira and Suzuki coupling of the components. The efficient triplet energy transfer with nearly quantitative quantum efficiency was shown to proceed upon excitation from the porphyrin unit to the anthracene unit (Figure 49) [103].

Especially dramatic influence is exerted by so-called push-pull, both donor and acceptor substituted porphyrins. Electron donor *N*,*N*-dimethylaniline and electron acceptor nitrobisthiophene-substituted acetylenes were attached to the opposite positions of the porphyrin via Sonogashira coupling (Figure 50). Such dipolar functionalized porphyrins possess considerable molecular hyperpolarizability and can be used for electro-optic applications [104,105].

Push-pull porphyrins have become the most efficient tetrapyrrole photosensitizers for dye-sensitized solar cells (DSSC). The dye with the donor diarylamino group and acceptor carboxyphenyl group, linked at the opposite *meso* positions with an ethyne bridge, outperformed all other porphyrins [106,107]. The synthetic strategy is similar to the examples given above. The Sonogashira coupling of *meso*-bromoporphyrin with trimethylsylylacetylene was carried out first, then the free *meso* position was brominated, followed by Buchwald amination with diarylamine, and after removing the trimethylsilyl protecting group, the second Sonogashira coupling with iodobenzoic acid was performed (Figure 51).

A large range of 5,15-bisalkynyl substituted porphyrin derivatives were obtained by Sonogashira coupling *meso*-dibromo-di(carboxyphenyl)porphyrins, which were shown to be suitable for the synthesis of surface-anchored MOF thin films [108].

The conjugated dimer and trimer with di- and triethynylbenzene bridges were obtained by the Sonogashira coupling of *meso*-ethynylporphyrin with di- and triiodobenzene. Whereas nonconjugated oligomers were obtained with tetrakis(4-iodophenyl)methane and tetrakis(4-iodophenyl)porphyrin (Figure 52) [109]. Different types of oligomers with diphenylacetylene bridges were obtained as a result of the Sonogashira coupling of *meso-*(4-ethynylphenyl)porphyrin with *meso-*(4-iodophenyl)porphyrin [110] and *meso-bis*(4-iodophenyl)porphyrin [111]. It is worth noting that the directly attached aryl group at *meso* position is not conjugated with the macrocycle because it turned almost perpendicular owing to the sterical interactions, and the dimers linked through the *meso*-phenyl [112], including the *meso*-diphenylacetylene bridge [111], are not conjugated [113].

The fully conjugated porphyrin dimer **131**, directly linked by acetylene at *meso* position was obtained using Stille-type coupling of bis-1,2-stannylacetylene with 5-halo-TrPP with a 43% yield (Figure 53) [39]. The electron absorption spectrum of the dimer **131** showed considerable changes compared to the monomer, which can be attributed to the extensive conjugation and strong interchromophore communication.

The DPP dimers and trimers bridged by ethyne linkers were obtained by the Sonogashira coupling of *meso*-ethynylporphyrin **132** and **134** with *meso*-bromo-porphyrin **111** (Figure 54) [92]. The dimers with diethynylarene bridges **137**–**140** were obtained by the Sonogashira coupling of partially protected *meso*-diethynylporphyrin **136** with diiodoarenes (Figure 55) [114]. The thiophene linker provided more conjugation than phenylene but less than anthracene, which allowed even more electronic communication than the simple butadiyne. All *meso*-arylporphyrin dimers with triple bond linkers possess high cross-sections of two-photon absorption, which makes them the most promising candidates for photosensitizers in two-photon-induced PDT. Efficient singlet oxygen generation was demonstrated both in one- and two-photon excitation of these dimers [115].

Further transformations of the acetylenylporphyrins include oxidative coupling, 1,3-dipolar cycloaddition, and nucleophilic addition. Glaser oxidative coupling of *meso*-acetylenylporphyrins was used to obtain porphyrin dimers as well as conjugates with other acetylenyl substituted compounds, but the yields of the oxidative cross-coupling are generally low due to the homocouplings. For example, 5,15-bisacetylenylporphine **141** was coupled with an excess of *meso*-acetylenyl-OEP, leading to the porphyrin triad **142** linked by butadiyne bridges with a 25% yield, along with a larger amount of the OEP dimer **143** formed by homocoupling (Figure 56) [95]. The conjugation in the triad led to the splitting of the Soret band of OEP into two main bands with clear maxima at 429 and 481 nm and a bathochromic shift of the Q band to 670 nm. Furthermore, a dramatic bathochromic shift of the Q-band by 70 nm was observed in pyridine compared to that in chloroform, probably due to the coordination of the pyridine molecule as an axial ligand onto the Ni cation, though Ni porphyrinates do not usually coordinate extra ligands.

The butadiyne-linked *meso*-diarylporphyrin dimer **148** functionalized with hydrophilic groups was obtained using a sequence of Sonogashira and Glaser-type oxidative coupling reactions. First, *meso*-di(trihexylsilylacetylenyl)-diarylporphyrin was obtained from easily accessible *meso*-dibromodiarylporphyrin **144** and trihexylsilylacetylene using the classical Sonogashira reaction. The partial deprotection of the triple bond with TBAF proceeded with a low 29% yield. The half-protected bisacetylenylporphyrin **145** was dimerized using oxidative coupling in modified conditions catalyzed by Pd(PPh_3_)_4_, CuI, and 1,4-benzoquinone as an oxidant. These conditions provided a high 94% yield of dimer **146** bridged by butadiyne linkers. The presence of two *meso-*trihexylsilylacetylenyl groups in the dimer allowed for further functionalization using Sonogashira coupling with iodo-substituted hydrophilic fragments (Figure 57) [116]. The obtained dyes possess red-shifted absorption bands in a region of 700–800 nm and a high two-photon absorption cross-section. These properties are important for application in PDT, and the porphyrin dimer dyes were studied as photosensitizers and were found to be more effective than the commercial drug Visudyne^®^ in two-photon PDT [40].

A series of bisporphyrins linked by bithiophene and butadiyne groups were obtained using oxidative cross-coupling of *meso*-acetylenyl-OEP with bisacetylenylbithiophene and oligomeric bithiophenes [97]. The coupling was carried out in the presence of copper(II) acetate in a mixture of pyridine and methanol. Yields of cross-coupled products were 15–20%, together with 20–30% yields of the diacetylene-bridged OEP dimer (Figure 58). The position of the hexyl substituents in bisthiophene determines the relative orientation of thiophene rings, which plays an important role in electronic communications between the two terminal OEP rings.

A series of *meso*-diarylporphyrin dimers linked by oligoacetylenes were obtained using Glaser coupling of *meso*-(oligoethynyl)porphyrins [94]. The similar coupling of the 5,10-diethynyl-15,20-diarylporphyrin **153** led to the formation of the square-shaped porphyrin tetramer **154** (Figure 59) [117]. The positions of both the Soret (503) and Q (659 nm) bands were bathochromically shifted by about 2900 cm^−1^ relative to the monomer but remained similar to those of the corresponding linear tetramer [118].

A square cyclic porphyrin dodecamer **158** with ethynyl linkers was obtained via the tetramerization of a T-shaped trimer **157** using Glaser oxidation coupling [119]. The synthesis of the trimer was based on the Sonogashira reaction of 5,10-diethynyl-15,20-porphyrin **156** with 5-iodo-15-bromo-10,20-diarylporphyrin **157** (Figure 60). The molecule was easily visualized using STM. The round-shaped octamer with butadiyne linkers **161** was synthesized via oxidative coupling of the 5,15-diethynyl-10,20-diarylporphyrin **159** using a template with palladium/copper catalysts and iodine as an oxidant to give the cyclooctamer **161** with a 14% yield (Figure 61) [120]. A similar cyclohexamer was obtained with a smaller template by oxidizing trimerization of the corresponding dimer [121]. Absorption and emission spectra showed that π-conjugation and interchromophore communication in the nanoring are stronger than in its linear analog and angled square-shaped macrocycles. The giant porphyrin cyclooligomers can be applied as artificial light-harvesting antennas. The similarity between these nanorings and the natural chlorophyll-based LH2 light-harvesting system [122] allows us to model the photosynthetic center with these artificial molecules [123]. Middle-sized, angled cycles like square porphyrin tetramer and dodecamer are host compounds that can coordinate suitable guest molecules, including fullerenes.

Most of the porphyrin dimers, trimers, and oligomers, linked by carbon-carbon triple bonds, were synthesized using either Glaser type oxidative coupling or Sonogashira coupling reactions. The considerable bathochromic shifts of the absorption and emission bands were observed for all multiporphyrin compounds compared to the precursor monomers. The Qy absorption bands of the porphyrin dimers are even longer in wavelength than those of the chlorin monomers, reaching up to 720 nm, making them promising photosensitizers for PDT and other optical applications such as optical sensors, NLO materials, etc. [124,125].

A copper-catalyzed 1,3-dipolar cycloaddition of azides to alkynes, called the “click” reaction, was used to create *meso*-1,2,3-triazole substituted porphyrin (Figure 62A). The porphyrin self-assembles to form a slipped cofacial dimer **164** by the coordination of the triazole nitrogen atom to the zinc center of a second porphyrin moiety (Figure 62B) [126].

The triazole group was also applied as a linker between porphyrin rings. Odobel obtained directly *meso*-*meso* triazole bridged dyads by the click reaction of Ni(II) and Zn(II) complexes of *meso*-ethynyltriarylporphyrin **166** with Ni(II) *meso*-azidotriarylporphyrin **165** (Figure 63) [127]. Both Ni-Ni and Ni-Zn dyads were obtained, but the yield of the heterometallic dyad **167**Zn was notably lower (18%) compared to the yield of the Ni-Ni dyad **167**Ni (41%). The reaction proceeded for quite a long time (50 h) in DMF with copper sulfate as a catalyst and ascorbic acid. Asymmetrical β-*meso*-triazole-linked dyad **169** was synthesized from nickel complexes of 5-ethynyl-10,20-diphenylporphyrin **132**Ni and β-azido-*meso-*tetraphenylporphyrin **168** [128]. The reaction was carried out in the same conditions but proceeded much faster, being completed for 1.5 h with a high 98% yield (Figure 63). It should be noted that in the case of the opposite reagent couple, namely 5-azido-10,20-diphenylporphyrin and β-ethynyl-*meso-*tetraphenylporphyrin, no reaction occurred under similar conditions.

The nucleophilic addition of alkynyl porphyrins to carbonyl compounds was used to prepare a series of porphyrin-dimer tertiary alcohols. Treatment of these alcohols with acid gave conjugated carbocations with three to nine carbon atoms bridging between the porphyrins (Figure 64). All these carbocations show strong absorption in the near-IR region between 1000 and 1800 nm [129].

## 5. Conclusions

Formyl, vinyl, and ethynyl are simple substituents that can easily be inserted into a tetrapyrrole macrocycle, providing suitable building blocks for the construction of porphyrin materials. The substitution at the *meso* position significantly affects the electron and optical properties of the porphyrins, and for this reason, it was the *meso*-derivatives that were considered. The reviewed works showed the rich potential of these synthons, opening the way to a variety of novel dyes with considerably modified properties that can be tuned by a choice of specific transformations of the starting building blocks. The products of such transformations are dyes for solar cells, light-harvesting antennas, photosensitizers for PDT, optical sensors, components for supramolecular ensembles, porous materials for storage and catalysis, etc. Particularly valuable are the biomedical applications of the tetrapyrrolic derivatives.

## Data Availability

Not applicable.

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
