# Peer review of "Meso-Formyl, Vinyl, and Ethynyl Porphyrins—Multipotent Synthons for Obtaining a Diverse Array of Functional Derivatives"

_molecules, 2023, doi:10.3390/molecules28155782_

Round 1

Reviewer 1 Report

Comments: 

Recommendation: accept in its current format 

The manuscript by Tyurin et al describes meso-formyl, vinyl, and ethynyl porphyrins -multipotent synthons for obtaining a diverse array of functional derivatives. This is a very informative and very important review article, the authors clearly presented the various functional derivatives of tetrapyrrole derivatives based on transformations of unsaturated carbon-oxygen and carbon-carbon bonds of the substituents at the meso position. Overall, the manuscript is written very well and a very interesting review for the readers, therefore the current version of the manuscript is acceptable for Molecule Journal.

Author Response

Authors thank the reviewer for the attention paid to this work.

Reviewer 2 Report

The submitted review paper is well prepared, comprehensive and valuable. The synthesis diagrams are of very good quality. The language is correct and understandable. I found no serious errors or inaccuracies. I propose to make/consider the following editorial changes:

1) Nitrogen atoms to which substituents are attached are written in italics. For example: is: N-bromosuccinimide, should be: N-bromosuccinimide.

2) We write the designation of the isomers in the bi-substituted benzene (o-, m- and p-) in italics. For example: is: di(p-acetylenylphenyl)anthracene, should be: di(p-acetylenylphenyl)anthracene.

3) The unit of hour(s) is "h" not "hrs". It is: 1.5 hrs, should be 1.5 hrs.

4) Rather, keywords are given in the singular. 

5) Please consider whether it is better to abbreviate ultraviolet-visible with "UV-vis" instead of "UV-Vis".

6) In the formula of fullerene C60, the number of carbon atoms rather should be in subscript.

7) Scheme captions should consistently end with a period. For example, the period is missing from the names of Schemes 20 and 21.

8) We do not give a space between the valency and the symbol, name, and this is generally the case in the manuscript. I noticed that spaces are inserted in l. 785 and 786.

9) Speaking of research funding. This is a review paper, so I would modify the funding notation. It cannot be: This research was funded...

10) Acknowledgements. "The authors acknowledge the prof. Ponomarev...". The definite preposition should be removed.

I commented on the language in the review (comments to authors).

Author Response

The manuscript was revised and corrected according to all the 10 points of  reviewer’s comments. Authors thank the reviewer for the attention paid to this work.

Reviewer 3 Report

In general, this is a well-written and exhaustive analysis of the synthesis and applications of tetrapyrrole compounds. The authors have done an outstanding job summarizing the various synthetic methods for producing these compounds as well as the diversified transformation pathways of the multipotent synthons. The structures and electronic and optical properties of compounds obtained through the methods under consideration are also thoroughly discussed. The authors have also provided an overview of the various applications of the obtained products, with a focus on biomedical applications. 

The review is well-organized and provides a concise summary of the subject. There are, however, a few areas that could be enhanced. For instance, the authors could elaborate on the benefits of using the reviewed synthetic strategy to obtain dyes with the desired characteristics. In addition, the authors could provide additional information on the compounds' potential applications, such as their use in solar cells, light-harvesting antennas, photosensitizers for photodynamic therapy (PDT), optical sensors, components for supramolecular ensembles, and porous materials for storage and catalysis.

I caution good/high-impact journals against accepting manuscripts, even if the work is well-written and discusses the findings of the literature. I have never submitted a journal review shorter than 30 to 35 pages (excluding the reference section). As a scientist, I do not want it to be simpler to publish review articles than research publications. Before it can be recommended for publication in a journal, this analysis must be substantially expanded. I deem it necessary to include a discussion of meso-phenyl derivatives in the review. In particular, meso-phenyl substituted N-Confused [10.1021/acs. jp.0c04779 10.1039/C7NJ01814B] and sulfonate phenyl [10.1016/j.ica.2022.120973] porphyrins.

In general, this is a well-written and exhaustive analysis of the synthesis and applications of tetrapyrrole compounds. The authors have done an outstanding job summarizing the various synthetic methods for producing these compounds as well as the diversified transformation pathways of the multipotent synthons. This review could be an invaluable resource for researchers in the field with a few minor modifications.

Author Response

Point 1:  The review is well-organized and provides a concise summary of the subject. There are, however, a few areas that could be enhanced. For instance, the authors could elaborate on the benefits of using the reviewed synthetic strategy to obtain dyes with the desired characteristics. In addition, the authors could provide additional information on the compounds' potential applications, such as their use in solar cells, light-harvesting antennas, photosensitizers for photodynamic therapy (PDT), optical sensors, components for supramolecular ensembles, and porous materials for storage and catalysis.

Response 1: The main objective of the review is to show the synthetic opportunities of porphyrin synthons, based on three related functions: formyl, vinyl and ethynyl. The benefits of their using to obtain dyes with the desired characteristics were briefly mentioned. The mentioned applications  include solar cells, light-harvesting antennas, photosensitizers for photodynamic therapy (PDT), optical sensors, components for supramolecular ensembles, and porous materials for storage and catalysis.

Point 2: I deem it necessary to include a discussion of meso-phenyl derivatives in the review. In particular, meso-phenyl substituted N-Confused [10.1021/acs. jp.0c04779 10.1039/C7NJ01814B] and sulfonate phenyl [10.1016/j.ica.2022.120973] porphyrins.

Response 2: The review includes meso-phenyl substituted porphyrins as well. N-Confused porphyrins could be included if appropriate meso formyl, vinyl, or ethynyl derivatives existed. To the best of our knowledge, no such derivatives have been reported yet. Sulfonate phenyl porphyrins are also beyond the review. The last reference [10.1016/j.ica.2022.120973] is not related to porphyrins, but to Benzoselenadiazoles.

Authors thank the reviewer for the attention paid to this work.